# *FvMYB79* Positively Regulates Strawberry Fruit Softening via Transcriptional Activation of *FvPME38*

**DOI:** 10.3390/ijms23010101

**Published:** 2021-12-22

**Authors:** Jianfa Cai, Xuelian Mo, Chenjin Wen, Zhen Gao, Xu Chen, Cheng Xue

**Affiliations:** 1College of Horticulture, Fujian Agriculture and Forestry University, Fuzhou 350002, China; caijf1168@163.com; 2Horticultural Plant Biology and Metabolomics Center, Haixia Institute of Science and Technology, Fujian Agriculture and Forestry University, Fuzhou 350002, China; m3185412018@163.com (X.M.); wen313563@163.com (C.W.); gaozhen0695@163.com (Z.G.); 3College of Life Sciences, Fujian Agriculture and Forestry University, Fuzhou 350002, China; 4State Key Laboratory of Crop Biology, College of Horticulture Science and Engineering, Shandong Agricultural University, Taian 271018, China

**Keywords:** fruit softening, strawberry, *FvMYB79*, *FvPME38*

## Abstract

Strawberry is a soft fruit with short postharvest life, due to a rapid loss of firmness. Pectin methylesterase (PME)-mediated cell wall remodeling is important to determine fruit firmness and softening. Previously, we have verified the essential role of *FvPME38* in regulation of PME-mediated strawberry fruit softening. However, the regulatory network involved in PME-mediated fruit softening is still largely unknown. Here, we identified an R2R3-type MYB transcription factor FvMYB79, which activates the expression level of *FvPME38*, thereby accelerating fruit softening. During fruit development, *FvMYB79* co-expressed with *FvPME38*, and this co-expression pattern was opposite to the change of fruit firmness in the fruit of ‘Ruegen’ which significantly decreased during fruit developmental stages and suddenly became very low after the color turning stage. Via transient transformation, FvMYB79 could significantly increase the transcriptional level of *FvPME38*, leading to a decrease of firmness and acceleration of fruit ripening. In addition, silencing of *FvMYB79* showed an insensitivity to ABA-induced fruit ripening, suggesting a possible involvement of *FvMYB79* in the ABA-dependent fruit softening process. Our findings suggest FvMYB79 acts as a novel regulator during strawberry ripening via transcriptional activation of *FvPME38*, which provides a novel mechanism for improvement of strawberry fruit firmness.

## 1. Introduction

Fruit softening is a major determinant of shelf life and commercial value. It is the consequence of multiple cellular processes, including changes in soluble sugar content, glycosylation of cellulose, and remodeling of cell wall structure. Fruit cell walls are highly enriched in pectin, occupying more than 50% of the wall materials [1]. Pectin is the most abundant class of macromolecule within the cell wall matrix [2]. Pectin is also abundant in the middle lamellae between primary cell walls, where it functions as major adhesions between cells. During fruit ripening, a range of pectin-degrading enzymes are secreted into the cell wall to modify pectin structure, resulting in pectin polymer degradation [3]. Pectin methylesterase (PME) catalyzes reactions on pectin de-methylesterification to generate carboxyl groups during the release of methanol and hydrogen ions. PME provides the hydrolysis substrate for polygalacturonase (PG), and functions synergistically with the PG enzyme to soften fruit [4]. Previous studies have found that PME activity significantly increased associated with the ripening and softening process of various fruits, including banana, mango, strawberry, and kiwi fruit [5,6,7,8]. During tomato fruit ripening, lower PME activity was associated with an increased content of methylesterification of pectin, a decreased level of total and chelator soluble polyuronides in cell walls, as well as a more soluble solids contain. Despite the decrease of PME enzyme activity in ripening fruits influences fruit pectin metabolism and soluble solids accumulation, it is unable to interfere with the ripening process [9]. During apple fruit ripening, the softening process is accompanied by the elevation of PME activity; meanwhile, apple fruit softening and *PME* transcript are both intensively influenced by ethylene and low temperature [10,11]. In our previous study, the strawberry PME, *FvPME38* regulated fruit softening at the maturity stage of strawberry fruit, and the expression level of *FvPME38* is positively regulated by abscisic acid (ABA) [12]. These results all indicated that PME-mediated cell wall modification is an essential process to control fruit rigidity and ripening. Although these studies help us to understand the biological function of *PME* in the control of fruit softening, the complex regulatory network involved in the PME-dependent module remains to be further explored.

Fruit development and maturation involve a complex network of plant hormones [13]. In strawberry, auxin is mainly produced in achenes, while abscisic acid (ABA), gibberellins, and ethylene are synthesized predominantly in receptacles [14]. The central regulators, auxin and ABA, act antagonistically to regulate strawberry ripening [14,15]. In the early stage of strawberry fruit development, high auxin content is necessary to maintain cell division and expansion [15,16,17,18]. In the later stage, ABA content gradually increases, accompanied by a series process of fruit ripening including softening, accumulation of flavonoid, sucrose and acid [19]. In developing strawberry fruit, ripening is dramatically promoted by ABA [20], whereas application of nordihydroguaiaretic acid (NDGA) (an inhibitor of ABA biosynthesis), significantly inhibits ripening [21]. Functional deficiency of ABA biosynthesis or ABA signal perception both attenuate strawberry fruit ripening [22]. Apparently, ABA plays a central role in control of strawberry fruit ripening.

Many transcription factors (TFs) have been identified in mediating ABA-dependent fruit ripening through binding the cis-elements of ABA-inducible genes [23]. Among them, MYB TFs represent one of a large gene family involved in the ABA-dependent regulatory process, which consists of the 1R-MYB, R2R3-MYB, 3R-MYB and 4R-MYB subfamilies according to the number of conserved MYB motifs at their N terminus [24]. Most MYB TFs in plants are R2R3-MYB TFs, participating in diverse aspects of plant development and plant response to environmental change [25]. FvMYB1 and FvMYB10, via the protein complex formation with FvbHLH3/33 and/or FvTTG1 (WD40 protein), a ternary MBW complex, are particularly involved in the regulation of flavonoid, proanthocyanidin, anthocyanin and sucrose accumulation during strawberry fruit development [26,27,28]. The extent of transcriptional and metabolic changes detected in *FvGAMYB*-RNAi silenced strawberry receptacles suggests that *FvGAMYB* regulates the biosynthesis of flavonoids and ABA [29]. *F**aMYB44.2* negatively regulates sucrose accumulation through repressing *FvSPS3* expression in strawberry fruit [30]. However, the mechanisms involved in MYBs-mediated fruit ripening remain unclear.

*Fragaria vesca*, the woodland strawberry, is emerging as a model for the cultivated octoploid strawberry due to its small and sequenced genome, diploidy (2*n* = 14, 240 Mb genome), small stature, ease of growth, short life cycle [31]. Among *F. vesca* accessions, ‘Hawaii 4’ develops white fruit, including white receptacles and white achenes, but, in contrast, ‘Ruegen’ bears red receptacles and red achenes [32]. In this study, we identified *FvMYB79*, an R2R3-MYB TF, participating in PME-mediated strawberry fruit ripening. Transient overexpression of *FvMYB79* promoted but silencing of *FvMYB79* delayed strawberry fruit ripening. The transcriptional regulation of *FvMYB79* on fruit ripening was correlated with the abundance of *FvPME38*. In addition, FvMYB79-FvPME38 regulatory module might be involved in ABA-dependent fruit softening. Our study provided further understanding on the transcriptional regulatory mechanisms underlying cell wall-mediated fruit softening.

## 2. Results

### 2.1. FvMYB79 Are Transcriptionally Correlated with FvPME38 during Strawberry Fruit Development

In our previous study, *FvPME38* and *FvPME39* regulated fruit softening at the maturity stage of strawberry fruit [12], while the upstream regulatory network of *FvPME38* and *FvPME39* was still unknown. To determine the complex regulatory network, we used the fruitENCODE database [33] to clarify DNA methylation, accessible chromatin and histone modifications of *FvPME38* and *FvPME39* genes during strawberry fruit development. Based on the speculation from 5 methylation cyanine (5mC) and DNaseI hypersensitive sites (DHS), from immature stage to ripe stage, the signal of 5mC significantly decreased, but DHS signal remarkably increased in the promoter region of the two *PME* genes, suggesting that the promoters of *FvPME38* and *FvPME39* are demethylated and became accessible only in the stage of ripening fruit (Figure 1A). In contrast, immature strawberry fruit contained higher hyper-H3K27me3 in the *FvPME38* loci (Figure 1A). To identify the possible upstream regulator of *FvPME38* and *FvPME39*, 1.5 kb promoter region was captured and analyzed by PlantCARE. Bioinformatics analysis indicated that the *cis*-elements in *FvPME38* and *FvPME39* promoters mainly contained the phytohormone-responsive motifs and TF binding sites (Figure 1B). These findings implied that the transcriptional level of *FvPME38* and *FvPME39* genes are controlled by diverse regulators including phytohormones and TFs.

In order to search for the upstream regulators of *FvPME38* and *FvPME39*, we obtained 609 TF members from bHLH, NAC, WRKY, MYB, bZIP and MADS gene family which have binding cis-elements in *FvPME38* and *FvPME39* promoters (Appendix A). Co-expression network analysis based on the transcriptome data from fruitENCODE database showed that 27 TFs were identified as positive or negative co-expressed members with *FvPME38* or *FvPME39* (Figure 2A and Appendix A). Their expression level were further examined in the five different stages of ‘Ruegen’ fruit by q-PCR assay. Among them, *FvWRKY55*, *FvMYB65*, *FvMYB79* and *FvNAC114* were expressed highest at ripening stage, which coincided with the expression pattern of *FvPME38* and *FvPME39* (Figure 2B). We further individually measured the *FvMYB79* transcript in achenes and receptacles of ‘Ruegen’ (red fruit) and ‘Hawaii-4’ (white fruit). The expression level of *FvMYB79* was both significantly up-regulated in achenes and receptacles at turning at the ripening stage which reached the highest point at the last stage of Ruegen fruit, but, in contrast, strongly decreases in both receptacles and achenes of Hawaii-4 from stage RS2 to RS4 (Figure 2C–F). Accompanying fruit development, the change tendency of *FvMYB79* transcript and fruit firmness was negatively correlated, implying that FvMYB79 may be involved in regulation of strawberry fruit firmness.

### 2.2. FvMYB79 Is an ER and Nucleus-Localized Transcriptional Factor

Sequence analysis of *FvMYB79* showed that it has a complete open reading frame (ORF) region (507 bp in length), which encodes a protein with 168 amino acids (GenBank accession number MN530978) with a calculated molecular mass of 19.38 kDa and isoelectric point of 10.44.

To predict the function of *FvMYB79*, a phylogenetic tree was constructed using the amino acid sequences of FvMYB79 and MYBs from other plant species, which are well known to control phenylpropanoid, anthocyanins and proanthocyanins biosynthesis (Figure 3A). Although FvMYB79 was clustered with the known anthocyanin- and proanthocyanin-associated MYB TFs, including production of anthocyanin pigment 1(PAP1), production of anthocyanin pigment 2 (PAP2), TRANSPARENT TESTA 2(TT2) from *Arabidopsis* [34,35,36], MdMYB10 from *Malus x domestica* [37], anthocyanin 2(AN2) from *Petunia hybrida* [38], VvMYBPA2 from Vitis vinifera [39], LjTT2a and LjTT2b from *Lotus japonicus* [40], MtMYB14 from *Medicago truncatula* [41], the bootstrap value of FvMYB79 was lower than 50 precent, indicating a potential different function of FvMYB79 (Figure 3A).

To examine the subcellular localization of the FvMYB79 protein, FvMYB79 was fused with N-terminal of green fluorescent protein (GFP) tag, driven by the CaMV 35S promoter. Visualization of FvMYB79-GFP by transient transformation in tobacco leaf cells showed that the FvMYB79-GFP signal was co-localized with the well-characterized endoplasmic reticulum (ER) marker tdTomato-HDEL, as well as the nucleus marker mRFP-NLS. Hence, FvMYB79 has a dual localization in both ER and nucleus (Figure 3B).

The amino acid sequence alignment analysis showed that FvMYB79 protein contains typical R2- and R3- MYB DNA-binding domains, and a highly conserved Lx2[R/K]x3Lx6Lx3R motif for interaction with other bHLH proteins [42,43,44] (Figure 3C). Previous studies showed that the EAR motif was able to convert a transcriptional activator into a strong repressor [45]. Furthermore, there isn’t ERF-associated amphiphilic repression domain (EAR) in the highly divergent C-terminal region of FvMYB79 (Figure 3C), suggesting that FvMYB79 may act as a transcriptional activator.

### 2.3. FvMYB79 Regulates Strawberry Fruit Softening via Transcriptional Activation of FvPME38

To further elucidate the functionality of *FvMYB79*, transient overexpression and RNA interference (RNAi) of *FvMYB79* were performed in ‘Ruegen’ fruit at 18 d after pollination (DAP). After infiltration, a dramatic delay of fruit ripening was found in *FvMYB79* RNAi fruits, but *FvMYB79* overexpressed fruits accelerated the ripening process (Figure 4A–C). Measurement of fruit rigidity showed that *FvMYB79* RNAi fruits has higher firmness but overexpression of *FvMYB79* reduced fruit firmness, indicating that *FvMYB79* influences fruit rigidity (Figure 4C,D).

To further study whether *FvMYB79* is involved in the regulation of fruit ripening, we examined the transcriptional level of well-known ripening marker genes, including pectin methylesterase (*PME*), expansin (*EXP*), beta-xylosidase (*XYL*), pectate lyase (*PL*), polygalacturonase (*PG*), and cellulose (*CEL*); the flavonoid biosynthesis genes chalcone synthase (*CHS*), chalconeisomerase (*CHI*), dihydroflavonol 4-reductase (*DFR*), UDP-glucose flavonoid 3-O-glycosyltransferase (*UFGT*) and phenylalanine ammonialyase (*PAL*). The transcript levels of *PME*, *EXP*, *PL*, *PG*, *CEL*, *CHS*, *CHI*, *MYB10*, *DFR*, and *UFGT* were significantly downregulated in the *FvMYB79* RNAi fruits, but upregulated in the overexpressed *FvMYB79* fruits, compared with the control (Figure 4E,F). However, in transient transformed fruits, we monitored the mild changes in expression of genes encoding regulatory proteins associated with fruit development and ripening, such as *RIF*, *MADS9*, *SHP*, *MYB44.2*, *SnRK2.6* and *MRLK47*, as well as hormonal, sugar and acid related genes (Appendix A) [30,46,47,48,49]. Based on the above expression assay, *FvMYB79* is possibly involved in the processes of anthocyanin biosynthesis and softening during fruit development. To understand if *FvMYB79* directly or indirectly influences anthocyanin biosynthesis, we overexpressed *FvMYB79* in ‘Hawaii-4’ fruits that showed white color throughout the whole fruit development. After seven days of infiltration, anthocyanin did not accumulate and transcripts of anthocyanin-related genes (*MYB1*, *bHLH3*, *bHLH33*, *TTG1* and *UFGT*) were not upregulated, even with mild reduction, in the fruit of overexpressed *FvMYB79* (Appendix A).

To test the regulatory network of *FvMYB79*, we firstly examined the interaction of FvMYB79 with potential partners in the MBW complex and other genes encoding regulatory proteins associated with fruit development and ripening. A Y2H assay showed that FvMYB79 was not able to interact with the proteins that are crucial for fruit ripening, including MADS9, SHP, SnRK2.6, RIF, TTG1, bHLH33 and bHLH3 (Appendix A). To search for the other downstream regulators of *FvMYB79*, some flavonoid and softening related genes were selected, of which expression levels were regulated by *FvMYB79* (Figure 4E,F). We cloned the 2.0 kb promoter sequences of them to perform dual-luciferase assay. The results indicated that FvMYB79 activated with *FvPME38* promoter in a specific manner (Figure 4G). To further investigate whether FvMYB79 could bind directly to the promoter of *FvPME38* yeast, one-hybrid assay was carried out. The self-activation activity of the promoter was screened with aureobasidin A (AbA), and the results showed that 30 ng·mL^−1^ AbA could inhibit the growth of yeast cells. Yeast one-hybrid screens indicates that yeast cells co-transformed with FvMYB79-effector and pFvPME38-report grew on SD/-Leu medium (with 0, 20× and 30× AbA), demonstrating that FvMYB79 could bind directly to the promoter of *FvPME38* (Figure 4H). Altogether, FvMYB79 plays a role in the regulation of strawberry fruit softening by controlling the transcriptional level of *FvPME38*.

### 2.4. FvMYB79 Might Be Involved in Abscisic Acid-Dependent Fruit Ripening Process

Endogenous ABA has been proved essentially for the onset of ripening process in strawberry fruit [50]. In our previous study, the expression level of *FvPME38* is positively regulated by ABA [12]. We thus speculated that FvMYB79-FvPME38 module might participate in ABA-dependent fruit ripening. We applied endogenous ABA or nordihydroguaiaretic acid (NDGA, an ABA inhibitor) on the ‘Ruegen’ fruits at the turning red stage, and then quantitatively examined fruit ripening by calculation of the percentages of different stages of fruits and measurement of fruit firmness. In comparison with the control group, exogenous ABA treatment significantly accelerated fruit ripening, displaying higher anthocyanin accumulation and reduced fruit firmness, whereas NDGA-treated fruits postponed anthocyanin accumulation and fruit softening (Figure 5A–C). Correspondingly, NDGA treatment significantly reduced and ABA treatment stimulated the transcript abundance of *FvMYB79* transcript (Figure 5D). In RNAi-*FvMYB79* fruits, the ABA effect on the promotion of fruit ripening was significantly delayed (Figure 5E–I). We further detected ABA content in *FvMYB79* RNAi fruits. The endogenous ABA content was dramatically reduced in *FvMYB79* RNAi fruits, in comparison to control (Figure 5J). We therefore postulated that *FvMYB79* may be involved in the regulation of ABA-dependent fruit ripening. These results also suggested that *FvMYB79* accelerates the fruit ripening process possibly mediated by the ABA signal.

## 3. Discussion

### 3.1. Identification of Regulator Combining Epigenetics, Transcriptomics and Co-Expression Analysis

Strawberry ripening is a highly coordinated program which leads to structural and bio-chemical changes, such as sudden receptacle softening and increase in the contents of sugars, anthocyanins, volatile compounds, and vitamins. The loss of firmness of receptacles is one of the main processes in the onset of fruit ripening due to the activity of cell wall remodeling enzymes. In our previous study, *FvPME38* and *FvPME39* regulated fruit softening in the onset of strawberry fruit ripening, and showed expression levels which were positively regulated by ABA [12]. Although these studies help to understand the biological function of *PME* when fruit softening, the complex regulatory network involved in the process remains to be further explored. Global transcriptome and co-expression network analyses are becoming increasingly used to infer gene function, and set common pathways and putative targets for TFs in many perennial plants which are difficult to use knockout or knockdown T-DNA mutant methods to verify gene function because of the long generation time [51,52]. In the present study, we provided a high-efficiency way to identify and predict the function of candidate genes genome-wide combining epigenetics, transcriptomics and co-expression analysis. Using known *FvPME38* and *FvPME39* genes as bait (target) genes, DNA methylation, accessible chromatin and histone modifications of those genes were further analysed to find out the sequence region of dramatic change of epigenetics signals during fruit development, and predicting cis-elements on the sequence region of those genes. Afterwards, all members of the probable TF gene family were identified or obtained from published database before. The combination of expression profiles from transcriptome data and physiological trait analysis paves the way to quickly select candidate genes correlated with the target trait. The findings from fruitENCODE [33] and PlantCARE [53] database showed that the transcriptional level of *FvPME38* and *FvPME39* genes are controlled by various factors, including phytohormones and TFs, during plant developmental processes. Further co-expression network analysis with transcriptome data showed that 27 TF genes from bHLH, NAC, WRKY, MYB, bZIP and MADS families were identified as positive or negative co-expression members with *FvPME38* or *FvPME39*. The expression level of candidate genes were further examined in the five different stages of ‘Ruegen’ fruit by q-PCR assay. *FvWRKY55*, *FvMYB65*, *FvMYB79* and *FvNAC114* were expressed highest at ripening stage, which coincided with the expression pattern of *FvPME38* and *FvPME39*. Among them, *FvNAC114*, namely *FaRIF* in previous study, was reported as a key regulator of strawberry fruit ripening [49]. In ‘Ruegen’ (red fruit) and ‘Hawaii-4’ (white fruit), the expression level of *FvMYB79* was both significantly up-regulated in achenes and receptacles after the ripening stage, which reached its highest at the last stage of Ruegen fruit, but, in contrast, strongly decreases in both receptacles and achenes of Hawaii-4 from stage RS2 to RS4. It suggests that, in the two genotypes, the function of *FvMYB79* was similar at the early stage of fruit ripening, but was different at the late stage. Finally, we functionally characterized a R2R3-type MYB, FvMYB79, which has not been described so far. Molecular and phenotypic analyses of transient *FvMYB79*-overpression and *FvMYB79*-RNAi fruits indicated that FvMYB79 promotes strawberry fruit softening through activating the transcriptional level of *FvPME38* (Figure 5K), but did not result in altered expression levels of hormonal, sugar and acid pathway related genes.

### 3.2. FvMYB79 Responses ABA Signal to Specially Promote Strawberry Fruit Softening

Different phytohormones have been established involving strawberry fruit growth and ripening processed [14]. Auxin is mainly produced in achenes, while abscisic acid (ABA), bioactive free base cytokinins, gibberellins, and ethylene are synthesized predominantly in receptacles [14]. Endogenous ABA content is low at early stages and gradually increases to control a series of fruit ripening programs, such as fruit softening and the accumulation of flavonoids, sucrose and acid [19]. Based on our results, we propose that ABA and *FvMYB79* may act in a positive regulatory feedback loop to promote strawberry fruit ripening. Firstly, transcript levels of *FvMYB79* are substantially diminished after application of NDGA, while significantly increased after ABA treatment. Exogenous application of ABA in fruits significantly promotes the ripening process, which could be delayed in *FvMYB79-*RNAi fruits. Those results indicated that ABA acts upstream of *FvMYB79*. Secondly, we found that ABA content significantly decreased in *FvMYB79* RNAi fruits when compared with control, revealing that *FvMYB79* acts upstream of ABA biosynthesis. This regulatory mechanism is similar with other TFs, such as *FaRIF*, which is involved in a feedback regulation loop with ABA to regulate the strawberry ripening process [49].

Combining analyses of molecular and phenotypic experiment, our results showed that *FvMYB79* activates the transcriptional level of *FvPME38* gene to specially promote strawberry fruit softening, but is not involved in the accumulation of flavonoid, sucrose and acid during strawberry fruit development. Transient overexpression or RNAi at the expression level of *FvMYB79* at the turning stage of ‘Rugen’ strawberry significantly changed transcriptional levels of softening and flavonoid-related genes, but the expression level of hormonal, sugar and acid related genes were not significantly changed in the transient transformed fruits. Previous studies showed that FvMYB10 and FvMYB1 interact with FvbHLH3/33 and/or FvTTG1 (WD40 protein) to form a ternary MYB-BHLH-WD40 (MBW) complex, which is a positive and negative regulator of anthocyanins and flavonoid accumulation, respectively [26,27,28]. However, unlike FvMYB10 and FvMYB1, FvMYB79 didn’t interact with FvTTG1 and/or FvbHLH3/33 (Appendix A). Furthermore, the overexpression of *FvMYB79* in ‘Hawaii-4’ fruits (white fruit) didn’t promote anthocyanin accumulation in the transient transformed fruits. Dual-luciferase assay and yeast one-hybrid assay analysis showed that FvMYB79 only significantly activates the promoter of *FvPME38*, but didn’t activate the promoter of flavonoid related genes, such as *MYB10*, *DFR* and *UFGT*. Altogether, these results demonstrated that *FvMYB79* plays an important role for the regulation of strawberry fruit ripening and softening by controlling the transcriptional level of *FvPME38*. In a previous study, we observed a dramatic delay of fruit ripening in *FvPME38* RNAi fruits, but overexpression of the *FvPME38* accelerated ripening process [12], which in accord with the phenotype of transient overexpression or RNAi of *FvMYB79* in ‘Ruegen’ fruits. The function of *FvPME38* was similar with other cell wall-modifying genes, such as *FvXTH9*, *FvXTH6* and *FaβGAL4*. Overexpression of these genes led to faster ripening by modification of the cell wall components in strawberry fruits [54,55]. Although *FvMYB79* acts upstream of ABA biosynthesis, we speculate that the promoting fruit ripening in *FvMYB79* overexpressed ‘Ruegen’ fruits was not caused by ABA signal. Because endogenous ABA content controls a series of fruit ripening programmed, such as fruit softening, accumulation of flavonoid, sucrose and acid [19], but *FvMYB79* only regulates fruit softening process without a change in the transcriptional level of hormonal, sucrose and acid related genes (Appendix A). A previous study showed that pectin demethylesterification by PME is the main source of plant-derived methanol [56], and methanol is crucial in control of plant growth and response to stresses [57,58,59]. Therefore, *FvMYB79* controls transcriptional level of *FvPME38*, which may determine the production of methanol via cell wall pectin demethylesterification to regulate strawberry fruit ripening.

## 4. Materials and Methods

### 4.1. Plant Materials

The seventh generation inbred lines of *F. vesca* accessions, namely Ruegen (Ru F7-4, red-fruited) and Hawaii 4 (PI551572, white-fruited) were used as wild-types in this study. The plants were grown in a greenhouse (16 h/8 h light conditions at 22 °C, at a relative humidity of 65%). The samples, used for RNA isolation, were frozen in liquid nitrogen immediately after collection and then stored at −80 °C.

### 4.2. Identification of Cis-Element on Promoters of FvPME Genes

DNA methylation, dynamic chromatin accessibility, and histone modification of *FvPME* genes loci in immature fruit and fully ripened strawberry fruit tissues were obtained from the fruitENCODE website [33]. Cis-elements on the 1.5 kb promoter sequences of *FvPME* genes were predicted by PlantCARE [53]. The position of the cis-elements were displayed using an online website: Gene Structure Display Server (GSDS) [60].

### 4.3. Co-Expression Network

RNA-seq data of strawberry was gained from the fruitENCODE website [33]. Pearson’s correlation coefficient (PCC) values were used as a measurement of expression similarity between gene pairs, and filtered with Excel software (parameter was set as >0.9). Visualizations of the data were carried out by Cytoscape software [61].

### 4.4. Gene Expression Analysis by qRT-PCR

cDNA, used for quantitative reverse transcription-PCR (q-PCR) analysis, was synthesized using one-step genomic DNA removal and a cDNA synthesis kit (Transgen, Beijing, China). Q-PCR was performed using the MonAmp^TM^ ChemoHS qPCR Mix (Monad, Wuhan, China). Primers were shown in Appendix A. The analysis was performed using three biological samples and three technical repeats. Relative expression levels of each gene were normalized to an internal control *Fvactin* and *FvGAPDH* by 2^−ΔΔCp^ algorithm [62].

### 4.5. Phylogenetics, Gene Structure and Motif Analyses

A phylogenetic tree was constructed using MEGA X [63] with neighbor-joining (NJ) criteria and verified using the maximum likelihood (ML) method, and 1000 bootstrap replicates were performed based on the multiple alignments of the full-length amino acid sequences using ClustalW [64].

### 4.6. Texture Analyses

TA.XT.plus Texture Analyser (Stable Micro Systems Ltd., Surrey, UK) along with the measuring probe P/5S (5 mm Spherical stainless steel, supplied with the Texture Analyser) were employed for texture determination. The system was equipped with texture profile analysis (TPA). Hardness was measured as the maximum penetration force (N) reached during tissue breakage. Measurable parameters were: pretest speed 1 mm·s^−1^; test speed 1 mm·s^−1^ penetrating distance of 5 mm into the fruit. The measurement was triggered automatically at 0.04 N.

### 4.7. Plasmid Construction

The primers used for plasmid construction are listed in Appendix A. The coding region of *FvMYB79* gene was amplified from the cDNA of ‘Ruegen’ strawberry using PrimerSTAR^®^ GXL DNA Polymerase (TaKaRa, Maebashi, Japan), sub-cloned into pDONR221, and then inserted into the binary vector pK7WG2D using Gateway^®^ Technology. For RNAi, the partial coding sequence of *FvMYB79* (301-504 bp) was sub-cloned into pDONR221, and then inserted into the binary vector pK7WIWG2D. The correct fusion constructs were transferred into *Agrobacterium tumefaciens* strain GV3101 by the freeze–thaw method.

### 4.8. Subcellular Localization

The coding region of *FvMYB79* gene was sub-cloned into pDONR221, and then inserted into the binary vector pGWB605 to form the fusion vector 35S:FvMYB79-GFP using Gateway^®^ Technology. The leaves of three-week-old *Nicotiana benthamiana* plants were infiltrated through their abaxial surfaces by *Agrobacterium* suspension (OD_600_ = 0.6). At 72 h post-infiltration, a fluorescence signal was visualized using a Zeiss LSM880 confocal microscope (Zeiss, Oberkochen, Germany).

### 4.9. Transient Transformation of Strawberry Fruit Flesh

Transiently transformed strawberry fruit flesh was carried out using agro-infiltration as previously described [65]. GV3101 strains which harbor *FvMYB79* overexpression or RNAi constructs was infiltrated into the ‘Ruegen’ or ‘Hawaii-4’ fruit flesh at 18 d after pollination (DAP) using syringes. Six plants or fruits were injected with each construct in triplicates. The transformed samples were placed in the dark at 22 °C overnight and then transferred to a phytotron (22 °C, 16-h of light and 8-h darkness) for seven days. The GFP signal of fruit was examined using the fluorescence dissecting stereomicroscope (Nikon, Tokyo, Japan). Pictures were taken and tissues were collected for downstream analysis.

### 4.10. Yeast Two-Hybrid Assay

A yeast two-hybrid (Y2H) assay was performed using the Matchmaker^®^ Gold Yeast Two-Hybrid System (Clontech, http://www.clontech.com/ (accessed on 10 September 2020). The coding region of *MYB79*, *MADS9*, *SHP*, *SnRK2.6*, *RIF*, *TTG1*, *bHLH3* and *bHLH33* were amplified using the primers listed in Appendix A and individually inserted into pGBKT7 bait vector or pGADT7 prey vector, respectively. The bait and prey constructs were co-transformed into yeast strain Y2H Glod according to the manufacturer’s instructions (Yeastmaker, Clontech, Mountain View, CA, USA). Y2H Glod cells were grown on SD/-Trp-Leu medium for at least three days. Positive colonies were confirmed by PCR and further grown on SD/-Trp-Leu-His-Ade medium.

### 4.11. Dual-Luciferase Reporter Assays

The dual luciferase assay was carried out according to previous report [66]. The promoter sequences were amplified from ‘Ruegen’ DNA and inserted into the pGreenII 0800-LUC vector to construct the reporter constructs. All these promoter constructs were individually transformed into *A. tumefaciens* strain GV3101 that contains the pSoup helper plasmid using the freeze-thaw method. *Agrobacterium* containing the effector vector or report vector were re-suspended separately with infiltration buffer (10 mM MgCl_2_; 200 μM acetosyringone; 10 mM MES, pH 5.5) and mixed in a ratio of 9:1 to a final concentration of OD_600_ 0.9-1. After that the mixed cultures incubated at 25 °C with 60 r.p.m. shaking for two hours before infiltration. Two-week-old *N. benthamiana* leaves were infiltrated with the mixed bacterial cultures using needleless syringes. Three days after infiltration, firefly luciferase (LUC) and renilla luciferase (REN) were assayed using dual-luciferase assay reagents (Promega, Madison, WI, USA).

### 4.12. Yeast One-Hybrid Assay Analysis

The CDS of *FvMYB79* and promoters of *FvPME38* were cloned and inserted into the pGADT7 vector and the pABAi vector, respectively. Then, the bait plasmid was linearized and integrated into Y1H Gold yeast (Clontech, http://www.clontech.com (accessed on 10 September 2020)), and tested the promoter autoactivation on SD/-Ura agar plate containing Aureobasidin A(AbA, 0, 100, 200, 300, 400, 500, 600, 800, 1000 ng/mL). Furthermore, the interaction between FvMYB79 and the promoters of *FvPME38* were detected on SD/-Leu medium with the optimum AbA concentration. The empty pGADT7 of the corresponding recombinant pABAi was used as the negative control.

### 4.13. Hormonal Treatment Assay of Strawberry Fruits

The concentration of hormone solution is in accordance with a previous study [14]. The concentration of stock solution was 50 mM and 5 mM in ethanol for abscisic acid (ABA) and nordihydro- guaiaretic acid (NDGA) (Aladdin, Beijing, China), respectively. The concentration of working solution for treatment were 1 mM for ABA and 100 μM for NDGA diluted in ddH_2_O. GV3101 strains which harbor *FvMYB79* RNAi constructs was infiltrated into the ‘Ruegen’ fruit flesh at 18 DAP using syringes. At the same time, about 20 μL ABA working solution was injected into the 18 DAP fruits.

### 4.14. Measuring Fruit ABA Content

Fruit Sample powder was dissolved in 900 μL ethyl acetate and 100 μL [^2^H6-ABA] (CSA number: 35671-08-0, OlChemIm, Olomouc, Czech Republic) internal standard (concentration: 100 ng/mL). Next, the samples were vortexed for 30 s and sonicated in 4 °C for 20 min. After centrifugation at 12,000 rpm for 3 min at 4 °C, the supernatant was obtained and then evaporated until dry using a vacuum concentrator (Labconco, America). The dried residues were diluted in 200 μL of 70% methanol at 4 °C and then vortexed for 10 s and sonicated in 4 °C for 5 min. After centrifugation at 12,000 rpm for 2 min at 4 °C, the supernatant was obtained and then filtered through a 0.22 mm PVDF membrane. ABA was separated and detected using high-performance liquid chromatography (Acquity UPLC, Waters, Milford, MA, USA) coupled with a triple quadrupole mass spectrometer (XEVO TQS MS, Waters, Milford, MA, USA).

### 4.15. Statistical Analysis

Statistical analysis was done by one-way ANOVA in the Graphpad prism 8.0 software (Graphpad Software, San Diego, CA, USA). Significant differences between groups were calculated using *p* < 0.05 in one-way ANOVA analysis, Tukey’s HSD post hoc test.

### 4.16. Accession Numbers

The GenBank accession numbers of MYB protein sequences are as follows: AmMYBMIXTA (CAA55725.1), AtMYB46 (AED91824), AtMYB58 (AEE29461), AtMYB63 (AEE36212), AtMYB85 (AEE84639), AtPAP1 (AEE33419), AtPAP2 (AEE34503), AtTT2 (AED93980), BoTT2 (ADV03957), EgMYB2 (CAE09057), LjTT2a (BAG12893.1), LjTT2b (BAG12894.2), MdMYB10 (ACQ45201), MtMYB14 (KEH32454.1), MtMYB5 (XP_003601609.1), PH4 (AAY51377), PhAN2 (AAF66728), PtMYB1 (ACA33851), PtMYB4 (AAQ62540), PtMYB8 (ABD60280), PtrMYB126 (XP_002303999), PtrMYB134 (XP_002308528), PtrMYB20 (KF148676), PtrMYB3 (AGT02395), PtrMYB6 (KF811025.1), VvMYB5a (AFG28177), VvMYB5b (AAX51291), VvMYBPA2 (ACK56131), ZmMYBC1 (AAK09327), ZmMYBPL (AAB67721).The CDS sequence of strawberry *FvMYB79* has been deposited in GenBank data library under the following accession number MN530978.

## 5. Conclusions

Our study identified an R2R3-type MYB transcription factor, FvMYB79, via epigenetics, transcriptomics and co-expression multi-analysis. We verified *FvMYB79* functions as a sensor of the ABA signal involved in the strawberry fruit softening process via transcriptional activation of *FvPME38*. Our findings shed light on the transcriptional regulatory mechanisms underlying hormone-mediated fruit softening to provide preliminary knowledge for improving strawberry fruit quality.

## Figures and Tables

**Figure 1 ijms-23-00101-f001:**
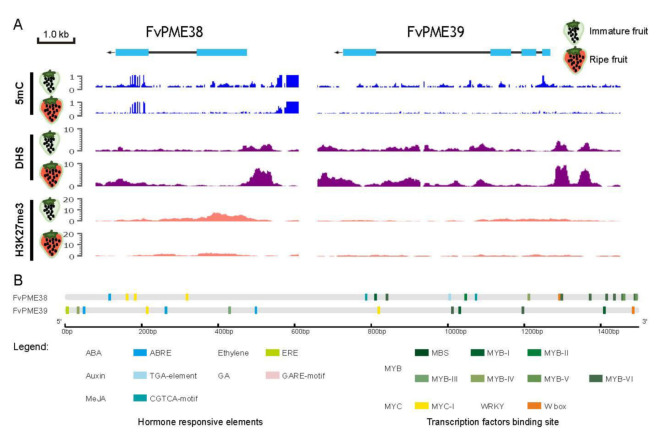
*FvPME38* and *FvPME39* genes are associated with tissue-specific epigenetic marks during strawberry fruit development. (**A**) DNA methylation, dynamic chromatin accessibility and histone modification of *FvPME38* and *FvPME39* genes in immature fruit and fully ripened strawberry fruit tissues. Individual data can be accessed on the fruitENCODE website. 5mC, 5 methylation cyanine; DHS, DNaseI hypersensitive sites; H3K27me3, H3 lysine-27 trimethylation; Green fruit, immature fruit; Red fruit, ripe fruit. (**B**) Putative cis-elements in the 1.5 kb promoter region of FvPME38 and FvPME39 genes.

**Figure 2 ijms-23-00101-f002:**
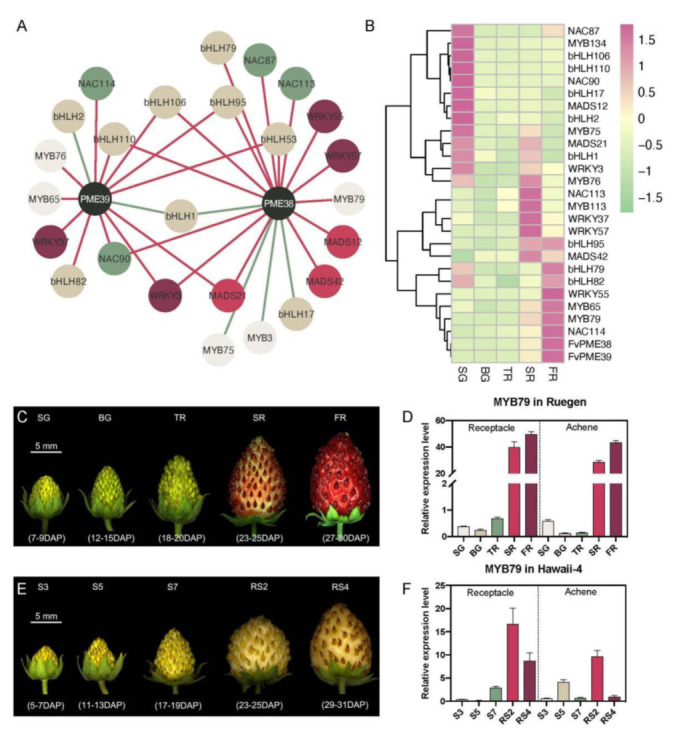
Co-expression network and expression profile of *FvMYB79*. (**A**) Co-expression network analysis of *FvPME38*, *FvPME39* and a series of transcription factor genes. Green line, negative correlation; red line, positive correlation. *P*-values were determined by two-tailed Student’s *t*-test assuming equal variances. (**B**) Relative transcription levels of candidate transcription factor genes were detected during ‘Ruegen’ fruit development by q-PCR. The data of each gene were centred and scaled in different stages via the heatmap package in R Studio Software, version 3.1.1. *P*-values were determined by two-tailed Student’s t-test assuming equal variances. (**C**,**E**) Five stages of ‘Ruegen’ and ‘Hawaii 4’ fruit development, respectively. ‘Ruegen’ fruit developmental stages: small green (SG), big green (BG), turning red (TR), start red (SR), and full red (FR). ‘Hawaii 4’ fruit developmental stages: S3, S5, S7, RS2 and RS4 were described previously [12,18]. (**D**,**F**) Spatial and temporal expression level of *FvMYB79* gene during ‘Ruegen’ and ‘Hawaii 4’ fruit development in achenes and receptacles. Error bars represent SD of three independent replicates.

**Figure 3 ijms-23-00101-f003:**
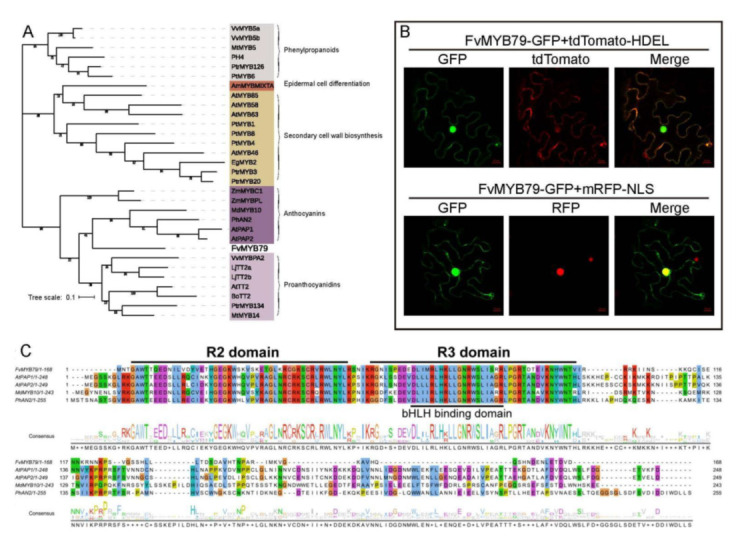
Phylogenetic analysis and subcellular localization of FvMYB79. (**A**) A phylogenetic tree was constructed by the maximum likelihood method of MEGA X software. Bootstrapping was performed 1000 times to obtain support values for each branch. (**B**) Subcellular localization of FvMYB79 was determined by FvMYB79-GFP fusion protein in tobacco leaf epidermal cells. (**C**) Alignment of the amino acid sequences of FvMYB79, AtPAP1, AtPAP2, MdMYB10 and PhAN2. R2 and R3 MYB-DNA-binding domains are underlined.

**Figure 4 ijms-23-00101-f004:**
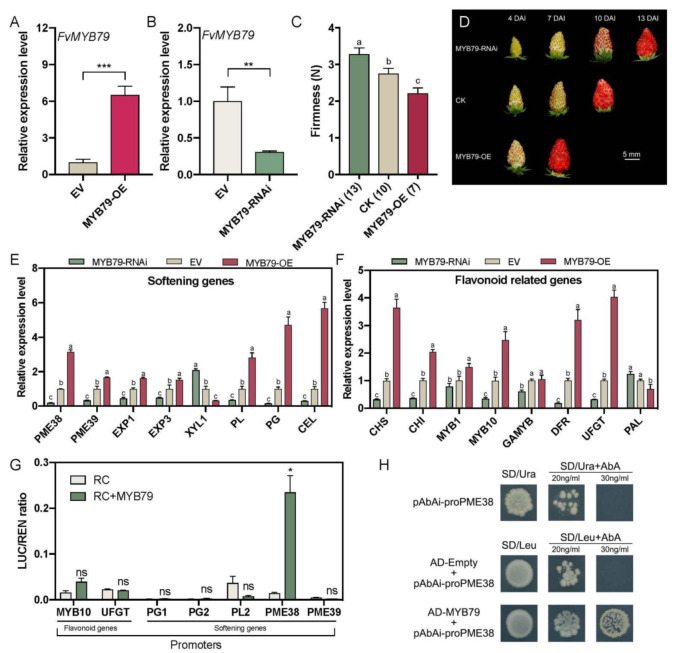
*FvMYB79* positively regulates fruit softening. (**A**,**B**) Q-PCR analysis of transcript levels for *FvMYB79* in overexpression and RNAi fruits at three days after injection. Error bars represent SD of three independent replicates. Asterisk indicates values that were determined by the *t*-test to be significantly different from the control (**, *p* < 0.01; ***, *p* < 0.001). (**C**) Firmness value of overexpression and RNAi fruits at same ripening stages after infiltration. The number in bracket of x-axis labels represents the number of days after injection. Error bars represent SD of 15 fruits. Letter indicates significant differences between groups (*p* < 0.05, one-way ANOVA, Tukey’s HSD post hoc test). (**D**) Phenotypes of fruits were agro-infiltrated with *FvMYB79* overexpression and RNAi constructs, respectively. DAI, day after infiltration; EV, empty vector; RNAi, RNA interference; OE, overexpression. (**E**,**F**) Expression assay of flavonoid and softening related genes in fruits of transient overexpression or silencing of *FvMYB79*. Relative expression levels of each gene were normalized to internal control *Fvactin*. Error bars represent SD of three independent replicates. PME, pectin methylesterase; EXP, expansin; XYL, beta-xylosidase; PL, pectate lyase; PG, polygalacturonase; CEL, cellulose; CHS, chalcone synthase; CHI, chalconeisomerase; DFR, dihydroflavonol 4-reductase; UFGT, UDP-glucose flavonoid 3-O-glycosyltransferase; PAL, phenylalanine ammonialyase. (**G**) Validation of activation effect of FvMYB79 on the promoter of flavonoid and softening related genes. Tobacco leaves were transfected with a reporter construct alone (RC) or together with the FvMYB79 effector construct (RC + MYB79). LUC, firefly luciferase; REN, renilla luciferase. Asterisk indicates values that were determined by the *t*-test to be significantly different from the control (*, *p* < 0.05). (**H**) The interaction between FvMYB79 and the *FvPME38* promoter. Autoactivation was tested on SD/-Ura in the presence of 20 and 30 ng/mL aureobasidin A (AbA). Physical interaction was determined on SD medium lacking Leu in the presence of 20 and 30 ng/mL AbA. The empty pGADT7 vector was applied as a negative control.

**Figure 5 ijms-23-00101-f005:**
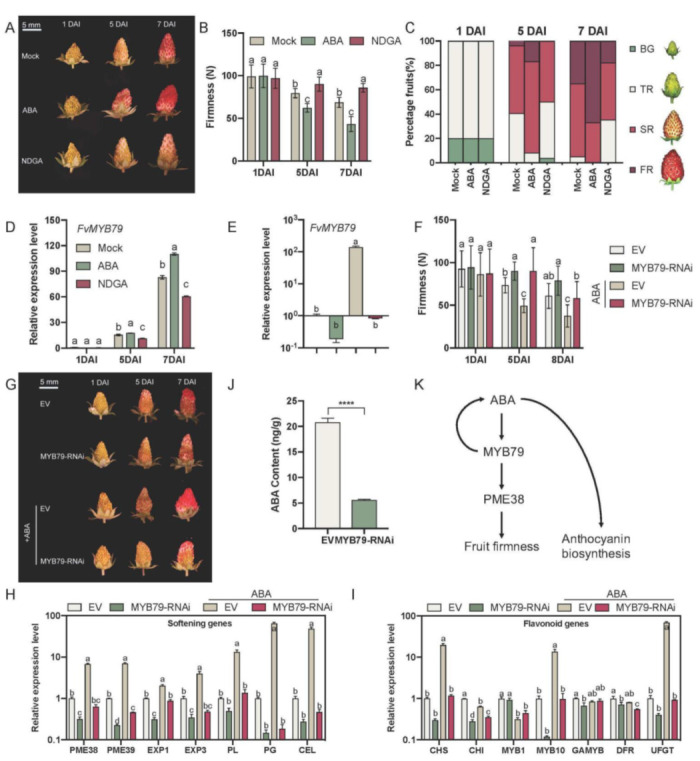
*FvMYB79* is involved in ABA-induced fruit softening. (**A**) Phenotypes of fruits were treated with ABA or NDGA (an ABA inhibitor). DAI, day after infiltration. (**B**) Firmness value of fruits after infiltration. The number in bracket of x-axis labels represents the number of days after injection. Error bars represent SD of 15 fruits. Letter indicates significant differences between groups (*p* < 0.05, one-way ANOVA, Tukey’s HSD post hoc test). (**C**) Percentage of fruits at each developmental/ripening stage for each time point. (**D**) Q-PCR analysis of *FvMYB79* expression level in strawberry fruits after five days treatment. Mock, 15 DAP fruits injected with water; NDGA, 15 DAP fruits injected with NDGA (100 μM). Error bars represent SD of three independent replicates. (**E**) Transcript levels for *FvMYB79* in control and *FvMYB79*-silenced fruits treated with ABA. Error bars represent the SD of three independent replicates. (**F**) The firmness value of fruits after different treatment. Error bars represent the SD of 15 fruits. (**G**) The Phenotypes of fruits were agro-infiltrated with empty vector or *FvMYB79* RNAi construct, and treated with ABA. DAI, day after infiltration; EV, empty vector; RNAi, RNA interference. (**H**,**I**) Expression assay of flavonoid and softening related genes in *FvMYB79*-silenced fruits treated with ABA. Relative expression levels of each gene were normalized to internal control *Fvactin*. Error bars represent SD of three independent replicates. (**J**) Endogenous ABA content was measured in fruits when *FvMYB79* was silenced. Error bars represent SD of three independent replicates. (**K**) The model of this study. In panel (**B**,**D**–**F**,**H**,**I**), letter indicates significant differences between groups (*p* < 0.05, one-way ANOVA, Tukey’s HSD post hoc test). In panel (**J**), asterisk indicates values that were determined by the t-test to be significantly different from the control (****, *p* < 0.0001).

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
