# Peer review of "FvMYB79 Positively Regulates Strawberry Fruit Softening via Transcriptional Activation of FvPME38"

_ijms, 2021, doi:10.3390/ijms23010101_

Round 1

Reviewer 1 Report

In this work the authors have demonstrated the role of the R2R3-type MYB transcription factor FvMYB79 in activating the expression of FvPME38 which positively affects fruit softening. Using a combination of tools such as DNA methylation status, quantification of gene expression, over-expression or RNAi of target genes, application of exogenous ABA the authors demonstrate the mechanism by which FvMYB79 affects fruit firmness during ripening. The work presented here will have potential impact on improving shelf-life in strawberries in the future. I have some minor comments to improve the current version.

Author Response

Thank you, we have made the correction.

Reviewer 2 Report

This work by Cai at al. provides and insightful analysis of the role of a MYB transcription factor in the ripening process, using strawberry as experimental system. The research has been carefully designed, conducted, analyzed and presented, an it is my suggestion that it can be published on this journal after minor revisions.

More in detail:

Figure 1 and relative section in Results and Discussion section:

  • As a suggestion, I recommend the Authors to report the full name of 5mC, DHS and H3K27m3 also in the legend, for quicker and simpler understanding of the image. 
  • More importantly, the levels of the 5mC marker have not been referenced in the Results and Discussion: they appear to me as similar for PME38 among the two ripening stages, while some differences are seen for PME39. The Authors should report this in the Results and comment its meaning in the Discussion.

Figure 2 and relative section in Results and Discussion section:

  • Hawaii-4 is a white variety. Is there a reason behind the choice of this variety? If so, the Authors could comment on it in their Results and Discussion paragraphs.
  • The expression of MYB79 strongly decreases in both receptacles and achenes of Hawaii-4 from stage RS2 and RS4. This does not happen in Ruegen, and maybe it would be interesting to mention/discuss it.

Interactions:

  • Y2H experiments were carefully carried on, yet no direct interactions were found among MYB79 and known ripening proteins. Can the authors discuss this results?

Conclusions:

  • The paper would benefit from a Conclusion paragraph (or at least a closing to the Discussion) where future perspectives are mentioned. For example, do these results about the functional characterization of MYB79 pave the way to a better understanding (and therefore exploitation) of the ripening pathway in commercial strawberry and / or related species? Are there genetic information linking alterations in this molecular pathway to improved ripening in certain varieties?

Minor typos:

  • References should be harmonized in the paper (see for example LINES 149-150-197)
  • LINE 215: is "programmed" wrong?
  • LINE 268: should "didn't involve" be changed to "is not involved"?
  • LINE 212: Remove "3. Discussion"

Author Response

Thank you, we have made the correction.
